# Synergistic Effect of Propidium Iodide and Small Molecule Antibiotics with the Antimicrobial Peptide Dendrimer G3KL against Gram-Negative Bacteria

**DOI:** 10.3390/molecules25235643

**Published:** 2020-11-30

**Authors:** Bee-Ha Gan, Xingguang Cai, Sacha Javor, Thilo Köhler, Jean-Louis Reymond

**Affiliations:** 1Department of Chemistry and Biochemistry, University of Bern, Freiestrasse 3, 3012 Bern, Switzerland; bee.gan@dcb.unibe.ch (B.-H.G.); xingguang.cai@dcb.unibe.ch (X.C.); sacha.javor@dcb.unibe.ch (S.J.); 2Department of Microbiology and Molecular Medicine, University of Geneva, 1211 Geneva, Switzerland; thilo.kohler@unige.ch; 3Service of Infectious Diseases, Geneva University Hospitals, 1211 Geneva, Switzerland

**Keywords:** antimicrobial peptides, dendrimers, membrane permeabilization, antibiotics, synergy

## Abstract

There is an urgent need to develop new antibiotics against multidrug-resistant bacteria. Many antimicrobial peptides (AMPs) are active against such bacteria and often act by destabilizing membranes, a mechanism that can also be used to permeabilize bacteria to other antibiotics, resulting in synergistic effects. We recently showed that **G3KL**, an AMP with a multibranched dendritic topology of the peptide chain, permeabilizes the inner and outer membranes of Gram-negative bacteria including multidrug-resistant strains, leading to efficient bacterial killing. Here, we show that permeabilization of the outer and inner membranes of *Pseudomonas aeruginosa* by **G3KL**, initially detected using the DNA-binding fluorogenic dye propidium iodide (**PI**), also leads to a synergistic effect between **G3KL** and **PI** in this bacterium. We also identify a synergistic effect between **G3KL** and six different antibiotics against the Gram-negative *Klebsiella pneumoniae*, against which **G3KL** is inactive.

## 1. Introduction

There is an urgent need to develop new antibiotics due to the increasing resistance of bacteria to the current antibiotics. Antibiotic resistance is particularly problematic with Gram-negative bacteria due to the presence of an additional outer membrane, which, in combination with multidrug efflux pumps, represents an efficient barrier against most antimicrobial compounds [1]. Many antimicrobial peptides (AMPs) show strong activities against multidrug-resistant Gram-negative bacteria [2,3,4,5,6,7,8,9]. Most AMPs act by directly disrupting the bacterial outer membrane and sometimes the inner membrane, and have therefore been investigated as permeabilizing agents for other antibiotics. These are typically antibiotics that act on intracellular targets on Gram-positive bacteria but are inactive on Gram-negative bacteria due to lack of cell penetration, leading to synergistic effects [10,11,12,13,14,15,16,17,18,19,20,21,22,23,24,25,26,27,28,29,30].

Herein, we report our investigation of possible synergistic effects between antibiotics and peptide dendrimer **G3KL** as a membrane permeabilizing agent (Figure 1). **G3KL** is an antimicrobial peptide dendrimer discovered by optimizing an initial combinatorial library hit [31] by sequence design, and exhibiting remarkable activity against Gram-negative bacteria such as *Pseudomonas aeruginosa, Acinetobacter baumannii*, and *Escherichia coli* including multidrug-resistant clinical isolates, but no activity against *Klebsiella pneumoniae* or methicillin-resistant *Staphylococcus aureus* (MRSA) [32,33]. **G3KL** selectively disrupts bacterial versus mammalian membrane models as evidenced by vesicle leakage assays [32], displaying pro-angiogenic properties in biological burn-wound bandages [34], anti-biofilm activity [35,36], low toxicity to mammalian cells (IC_50_ ~1000 μg/mL) [37], and low propensity to resistance development [38]. Our study was motivated by our recent observation that **G3KL** acts as a rapid membrane permeabilizer and strong membrane disruptor of bacterial cells. In this study, we show that **G3KL** destabilizes the LPS (lipopolysaccharide) layer, disrupts the outer and the inner membranes, interacts with DNA, and accumulates in Gram-negative bacteria up to an amount of dendrimer corresponding to 10% of the bacterial weight [37].

## 2. Results and Discussion

### 2.1. Membrane Permeabilization and Synergy with Propidium Iodide

We previously showed that **G3KL** permeabilizes the outer and inner membranes of *P. aeruginosa* cells using fluorescence microscopy and propidium iodide (**PI**), a fluorogenic DNA-binding dye, which is impermeable to intact cell membranes [37]. Permeabilization of bacterial membranes by **G3KL** might enable entry of potent antibacterial compounds at sub-inhibitory concentrations of **G3KL** and the cytotoxic compound. To test the feasibility of this approach and possible synergy between antimicrobials and **G3KL**, we used the classical checkerboard assay and stained live bacteria with 3-(4,5-dimethylthiazol-2-yl)-2,5-diphenyltetrazolium (MTT) (Figure 2) [32,39,40]. Synergy is a positive interaction in which the effect of the combined drugs is greater than when they are used alone. An additive effect indicates that the effects of the drugs used together are the same as when used independently. An indifferent effect is observed when the combination of the drugs is as efficient as the most potent drug alone. Antagonism is a negative effect observed when the combined effect of the drugs is significantly less than expected [41,42]. Interpretation of the fractional inhibitory concentration index (FIC_i_) was followed as described by Park et al. [43]. We considered a synergistic effect for FIC_i_ < 0.5; partial synergy for 0.5 ≤ FIC_i_ < 1; additive for FIC_i_ = 1; indifferent for 1 < FIC_i_ < 4; antagonism for FIC_i_ ≥ 4.

We investigated *P. aeruginosa*, *A. baumannii*, *E. coli*, and *K. pneumoniae* as Gram-negative bacteria and MRSA as Gram-positive bacterium. We first tested whether the permeabilization effect of **G3KL** observed with **PI** might result in a synergistic effect with this dye, which binds to DNA and therefore should interfere with bacterial growth [44]. When used alone, **PI** was essentially inactive against Gram-negative bacteria but showed moderate activity against MRSA (methicillin-resistant *Staphylococcus aureus*) (Table 1). In the checkerboard assay, we observed a partial synergistic effect with **G3KL** in *P. aeruginosa* (FIC_i_ = 0.5, Figure 2A), in line with our previous microscopy studies [37]. Partial synergy also occurred with *E. coli* (FIC_i_ = 0.6, Appendix A) and *A. baumannii* (FIC_i_ < 0.53, Appendix A). These data suggest that **G3KL** permeabilizes the bacterial membranes below its MIC value to facilitate the entry of **PI**.

In the case of *K. pneumoniae*, neither **PI** nor **G3KL** nor their combination showed any activity (Appendix A). As **G3KL** permeabilizes *K. pneumoniae* toward other antibiotics (see below), we interpret these data as an indication that **PI** is inactive against this bacterium at the highest concentration used, even in the presence of membrane permeabilization. The effect between **PI** and **G3KL** was indifferent in the case of MRSA, indicating that **G3KL**, which is inactive against this bacterium, also does not significantly increase the activity of **PI** against this bacterium (Appendix A).

By comparison, polymyxin B (**PMB**), a well-known membrane-disrupting natural antimicrobial cyclic peptide [45,46], showed partial synergistic effects with **PI** in *P. aeruginosa* (FIC_i_ = 0.6, Appendix A), *E. coli* (FIC_i_ = 0.6, Appendix A), *A. baumannii* (FIC_i_ < 0.8, Appendix A), and MRSA (FIC_i_ = 0.8, Appendix A), but showed a surprising antagonistic effect in the case of *K. pneumoniae* (FIC_i_ > 4, Appendix A), which is difficult to rationalize as **PMB** is known to permeabilize this bacterium [45].

### 2.2. **G3KL** Synergizes with Small Molecule Antibiotics against K. pneumoniae

We next conducted a broader survey to test if membrane permeabilization by **G3KL** might enable synergistic effects with classical antibiotics using the checkerboard assay with *P. aeruginosa* and *K. pneumoniae* as Gram-negative bacteria and MRSA as Gram-positive bacterium. We tested vancomycin [47], erythromycin [48], and trimethoprim [49], which are active against Gram-positive bacteria but inactive against Gram-negative bacteria due to inefficient uptake. We also tested novobiocin, which is most active against Gram-positive bacteria but shows activity against some Gram-negative pathogenic strains [50,51], as well as the broad-spectrum antibiotics ciprofloxacin [52], chloramphenicol [53], gentamicin [54], azithromycin [55], sulfamethoxazole, and ampicillin [56].

In *P. aeruginosa*, the combination of **G3KL** with ampicillin did not lead to synergy (Appendix A), as expected, due to the presence of the AmpC β-lactamase and the MexAB-OprM efflux pump in this bacterium [57]. We observed partial synergy of **G3KL** in combination with vancomycin (FIC_i_ = 0.6), chloramphenicol (FIC_i_ = 0.6), and azithromycin (FIC_i_ = 0.5), in line with the partial synergy observed with **PI** (Appendix A). Combination of **G3KL** with erythromycin or sulfamethoxazole led to an additive effect (FIC_i_ = 1, Appendix A), indicating that, in both cases, **G3KL** and the antibiotic act independently on their target without interfering with each other. An indifferent effect was observed between **G3KL** and novobiocin, ciprofloxacin, gentamicin, and trimethoprim (Appendix A). In general, the absence of synergy reflects that the concentration of **G3KL** necessary for membrane permeabilization (1–2 μg/mL) is very close to its MIC value when used alone (4–8 μg/mL). Nevertheless, **G3KL** was shown to kill *P. aeruginosa* within two hours by disrupting the outer membrane, the inner membrane, and ultimately by complexing with some negatively charged intracellular components, which probably includes DNA and proteins [37]. Thus, it is possible that **G3KL** impairs the binding of small molecule drugs to such targets, which would also contribute to the absence of synergistic effects.

In the case of *K. pneumoniae*, we observed a strong synergy between **G3KL** and vancomycin (FIC_i_ < 0.3, Figure 2B), erythromycin (FIC_i_ < 0.3), novobiocin (FIC_i_ < 0.2), chloramphenicol (FIC_i_ < 0.5), azithromycin (FIC_i_ < 0.4), and trimethoprim (FIC_i_ < 0.5), and partial synergy with the broad-spectrum antibiotic gentamicin (FIC_i_ < 0.6) (Appendix A, Table 2). The synergistic effects between **G3KL** and vancomycin, erythromycin, or trimethoprim are particularly striking because these compounds were inactive against *K. pneumoniae* when used alone. These synergistic effects suggest that **G3KL** was able to permeabilize *K. pneumoniae* cells even if it was not active against the bacterium, in line with that close derivatives of **G3KL** showing significant activity against *K. pneumoniae* [58,59,60]. Synergies have been reported previously between polymyxin B/colistin and small molecule antibiotics in *K. pneumoniae* [10,11,12,13,14,30,61]. Synergistic effects involving a permeabilizing but inactive compound have been previously observed with polymyxin B nonapeptide (PMBN), an inactive polymyxin B derivative lacking the fatty acyl chain [11,62].

**G3KL** did not increase the activity of ampicillin and sulfamethoxazole, which were inactive against *K. pneumoniae* when used alone (Appendix A). In the case of ampicillin, the antibiotic is probably degraded by the naturally-occurring β-lactamase in *K. pneumoniae* NCTC 418, rendering permeabilization inefficient [63]. **G3KL** also did not significantly increase the activity of ciprofloxacin, which is very active and whose activity is not limited by uptake (Appendix A).

Finally, we tested the permeabilization effect of **G3KL** in a Gram-positive MRSA strain. The results showed a very weak permeabilization effect, in line with **G3KL** being inactive against this bacterium. We observed a weak synergy between **G3KL** with erythromycin (FIC_i_ < 1), ampicillin (FIC_i_ < 1), and azithromycin (FIC_i_ < 0.8) (Appendix A) and an indifferent effect with all the other antibiotics (Appendix A), suggesting that **G3KL** is unable to pass the peptidoglycan layer and reach the membrane, and therefore cannot permeabilize MRSA for the uptake of small molecule drugs.

To confirm the synergistic effect observed in the checkerboard assay, we performed time-kill experiments for the combinations of **G3KL** with vancomycin (FIC_i_ < 0.3), erythromycin (FIC_i_ < 0.3), novobiocin (FIC_i_ < 0.2), and trimethoprim (FIC_i_ < 0.5). Killing kinetics on *K. pneumoniae* at an initial inoculum of ~10^6^ CFU/mL showed that the pairs **G3KL**/vancomycin (32 μg/mL/32 μg/mL) and **G3KL**/trimethoprim (32 μg/mL/16 μg/mL) effectively killed the bacteria after 4 h, and **G3KL**/erythromycin (32 μg/mL/16 μg/mL) after 8 h, all below the MIC level (Figure 3). The pair **G3KL**/novobiocin did not show a reduction in bacterial burden but exhibited a growth-inhibiting activity. The low level of surviving bacteria might not have been detected in the checkerboard assay. Similar growth inhibition was observed for novobiocin when used alone at 2 × MIC (16 μg/mL) against *K. pneumoniae* (Figure 3 and Appendix A). This effect was also observed in previous studies against *E. coli*, which suggested that novobiocin generally inhibits cell division and induces slower cell growth [50,64]. Note that **G3KL**, vancomycin, trimethoprim, erythromycin, and novobiocin when used alone at the same concentration as in combination showed no effect on bacterial killing.

## 3. Materials and Methods

### 3.1. Compounds

Peptide dendrimer **G3KL** was synthesized by solid-phase peptide synthesis and purified as described earlier [32]. Vancomycin, ampicillin, novobiocin, azithromycin, sulfamethoxazole, and trimethoprim were purchased from Sigma Aldrich (Buchs, Switzerland), erythromycin and ciprofloxacin were purchased from Acros Organics (Geel, Belgium), and chloramphenicol and gentamicin were purchased from AppliChem (Darmstadt, Germany). All compounds were conditioned as 8 or 10 mg/mL stock solutions in water (**G3KL**, ampicillin, novobiocin, chloramphenicol, and gentamicin), 1% acetic acid (ciprofloxacin), and DMSO (erythromycin, azithromycin, sulfamethoxazole, and trimethoprim).

### 3.2. Broth Microdilution Method

Antimicrobial activity was assayed against *P. aeruginosa* (PAO1), *A. baumannii* (ATCC19606), *E. coli* W3110 (TE823), *K. pneumoniae* (NCTC418), and methicillin-resistant *S. aureus* (COL). The minimal inhibitory concentration (MIC) was determined by using the broth microdilution method. A colony of bacteria was picked and grown in a Luria-Bertani (LB, Sigma Aldrich, Buchs, Switzerland) medium overnight at 37 °C. Stock solutions of 1 mg/mL of the samples were prepared in sterilized Milli-Q water and diluted to the starting concentration of 128 μg/mL in 300 μL Mueller Hinton (MH) medium. The diluted samples were added to the first well of the 96-well microtiter plate (TPP, untreated, Faust Laborbedarf, AG, Schaffhausen, Switzerland) and diluted serially by ½. Bacteria were quantified by measuring the optical density at 600 nm and diluted to OD_600_ of 0.022 in MH medium. We used 4 μL of the diluted bacterial solution to inoculate the sample solutions (150 μL) with a final inoculation of about 5 × 10^5^ CFU/mL. The plates were then incubated at 37 °C for 18 h. For each assay, sterility (broth only) and growth control (broth with bacterial inoculum, without antibiotics) were checked with two columns in the plate. The next day, 15 μL of MTT (1 mg/mL stock solution) (Sigma Aldrich, Buchs, Switzerland) was added to each well of the plate. The MIC was defined as the lowest concentration of the peptide dendrimer with a colorless well indicating no bacterial growth.

### 3.3. Checkerboard Assay

To verify the activity of two drugs in combination, the checkerboard method was used to determine the MICs for each antibiotic alone and in combination. Both **G3KL** [37] and a paired small molecule (propidium iodide, sulfamethoxazole, chloramphenicol, novobiocin, azithromycin, erythromycin, ciprofloxacin, gentamicin, ampicillin, and trimethoprim) were diluted by 1/2 in a 96 well plate. Stock solutions of 1 mg/mL of the antibiotics were prepared in sterilized Milli-Q water and diluted to the starting concentration with 2–4 × MIC of the corresponding compounds in 300 μL Mueller Hinton (MH) medium. For propidium iodide, the starting concentration was 1000 μg/mL. For checkerboard assay testing the combinations of **PI** with **G3KL**, **PMB**, and ciprofloxacin, **PI** was diluted across the rows and **G3KL**, **PMB**, and ciprofloxacin across the columns. For checkerboard assay testing the combinations of **G3KL** with small-molecule antibiotics, **G3KL** was diluted across the rows and the small-molecule antibiotics across the columns. We then added 75 μL containing 2× the final concentration of **G3KL**, **PMB**, and ciprofloxacin to each well containing the respective drug (**PI** and the small-molecule drugs) to be tested in combination, across the columns and down the rows, resulting in a checkerboard of 150 μL final volume with the final wells containing only **G3KL** or the antibiotics.

Similar to broth microdilution, bacteria were quantified by measuring the optical density at 600 nm and diluted to OD_600_ of 0.022 in an MH medium. We used 4 μL of the diluted bacterial solution to inoculate the sample solutions (150 μL) with a final inoculation of about 5 × 10^5^ CFU/mL. The plates were then incubated at 37 °C for 18 h. The next day, 15 μL of MTT (1 mg/mL stock solution) was added to each well of the plate so that MIC and MIC_comb_ were defined as the lowest concentration of the peptide dendrimer and/or antibiotics with a colorless well indicating no bacterial growth. The FIC of each antibiotic was calculated as follows:FIC_A_ (MIC_Acomb_/MIC_A_) + FIC_B_ (MIC_Bcomb_/MIC_B_) = ΣFIC = FICindex (FIC_i_).

A synergistic interaction is defined by an FIC index of <0.5, partial synergy is defined by an FIC index of ≥0.5 and <1, an additive interaction by an FIC index of 1.0, indifferent by an FIC of >1 and <4, and antagonism is defined by an FIC of ≥4 [43].

### 3.4. Time-Kill Assay

Time-kill kinetics against *K. pneumoniae* (NCTC418) were performed on **G3KL** (32 µg/mL), vancomycin (32 μg/mL), erythromycin (16 μg/mL), novobiocin (2 μg/mL), **G3KL**/vancomycin (32/32 μg/mL), **G3KL**/trimethoprim (32/16 μg/mL), and **G3KL**/erythromycin (32/16 μg/mL) and using 2× the concentrations indicated by the checkerboard assay. Untreated bacteria at 1 × 10^6^ CFU/mL were used as a growth control.

A single colony of *K. pneumoniae* (NCTC418) was picked and grown overnight with shaking (180 rpm) in a 5 mL LB medium (Sigma Aldrich, Buchs, Switzerland) at 37 °C. The overnight bacterial culture was diluted to OD_600_ 0.002 (2 × 10^6^ CFU/mL) in fresh MH (Sigma Aldrich, Buchs, Switzerland) medium. Stock solutions of **G3KL** and antibiotics (8 mg/mL) were prepared in sterilized Milli-Q water and diluted to two-times more concentrated than the required concentration in the fresh MH (Sigma Aldrich, Buchs, Switzerland) medium. **G3KL** and antibiotic were pre-mixed when needed. We mixed 100 µL of the adjusted bacteria and 100 µL samples in a 96-well microtiter plate (TPP, untreated, Corning Incorporated, Kennebunk, ME, USA) and the time-kill kinetics started at the moment of mixing. Ninety-six-well microtiter plates were incubated at 37 °C under shaking (180 rpm). Bacterial growth was quantified at 0, 0.5, 1, 2, 3, 4, 5, and 6 h, **G3KL**/erythromycin (32/16 μg/mL) were also quantified at 7 and 8 h. The quantification was performed by plating 10-fold dilutions of a sample in sterilized 0.9% NaCl on LB agar plates. LB agar plates were incubated at 37 °C for 10 h and the number of individual colonies was counted at each time-point. The assay was performed in triplicate.

## 4. Conclusions

The experiments above showed that the membrane permeabilizing effects of antimicrobial peptide dendrimer **G3KL** can be exploited to obtain synergistic effects with other substances. **G3KL** showed only weak or no synergy with small-molecule antibiotics in the case of *P. aeruginosa*, a bacterium against which the dendrimer is very active, probably because the concentration of **G3KL** necessary to permeabilize the membrane is very close to the MIC value. **G3KL** showed very significant synergistic effects when tested against *K. pneumoniae*, against which the dendrimer is inactive when used alone, revealing that **G3KL** is capable of permeabilizing the membrane even if it does not show activity. The effect is particularly striking in combination with vancomycin, erythromycin, or trimethoprim because these antibiotics are inactive against *K. pneumoniae* when used alone. However, our synergistic study showed no effects against MRSA, indicating that **G3KL** is not only inactive against this Gram-positive bacterium, but also does not significantly permeabilize its membrane.

## Figures and Tables

**Figure 1 molecules-25-05643-f001:**
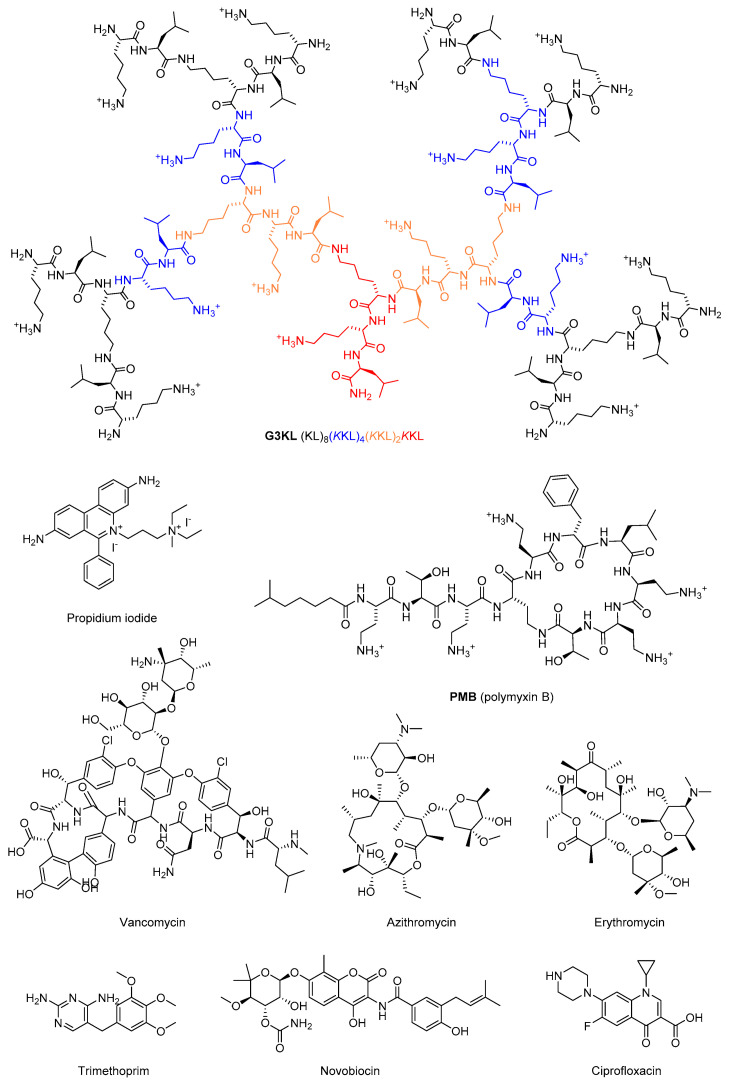
Structural formula of antimicrobial compounds used in this study.

**Figure 2 molecules-25-05643-f002:**
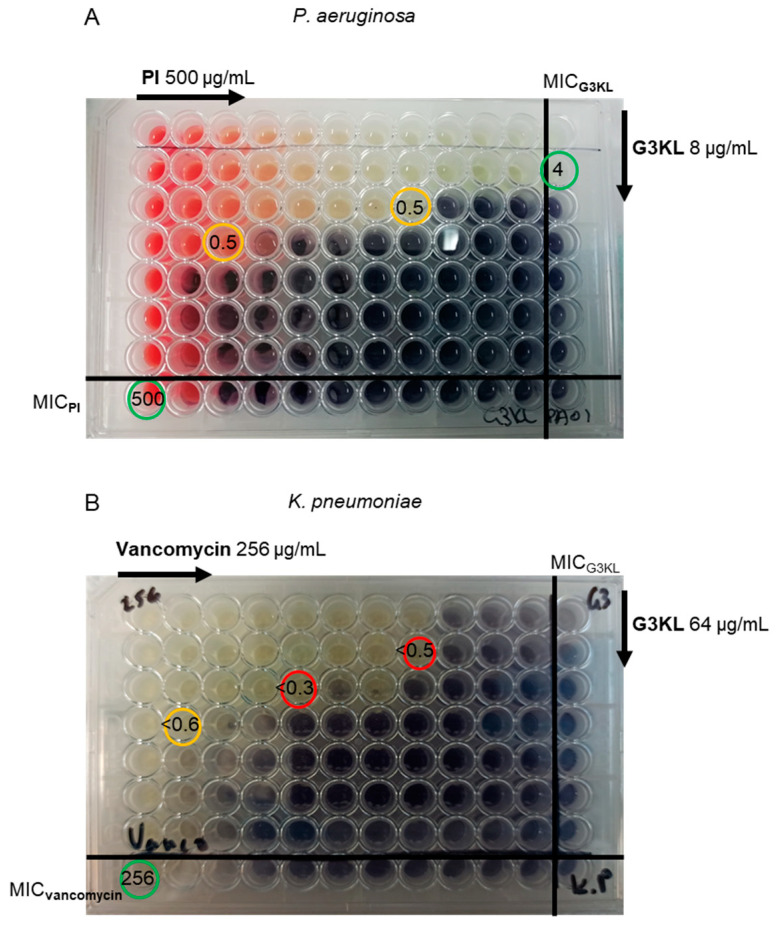
Checkerboard microtiter plate assay testing the combination of **G3KL** with **PI** in *P. aeruginosa* PAO1 (**A**) and vancomycin with **G3KL** in *K. pneumoniae* NCTC 418. (**B**) 2D two-fold serial dilutions were performed starting with 500 µg/mL of **PI**, 256 µg/mL of vancomycin, and 8 or 64 µg/mL of **G3KL**. The viability of the bacteria was revealed after the addition of 3-(4,5-dimethylthiazol-2-yl)-2,5-diphenyltetrazolium (MTT) dye (black wells). Red circle: FIC_i_ of synergistic effect; yellow circles: FIC_i_ of partial synergy; green circles: MIC values of **G3KL**, **PI**, and vancomycin.

**Figure 3 molecules-25-05643-f003:**
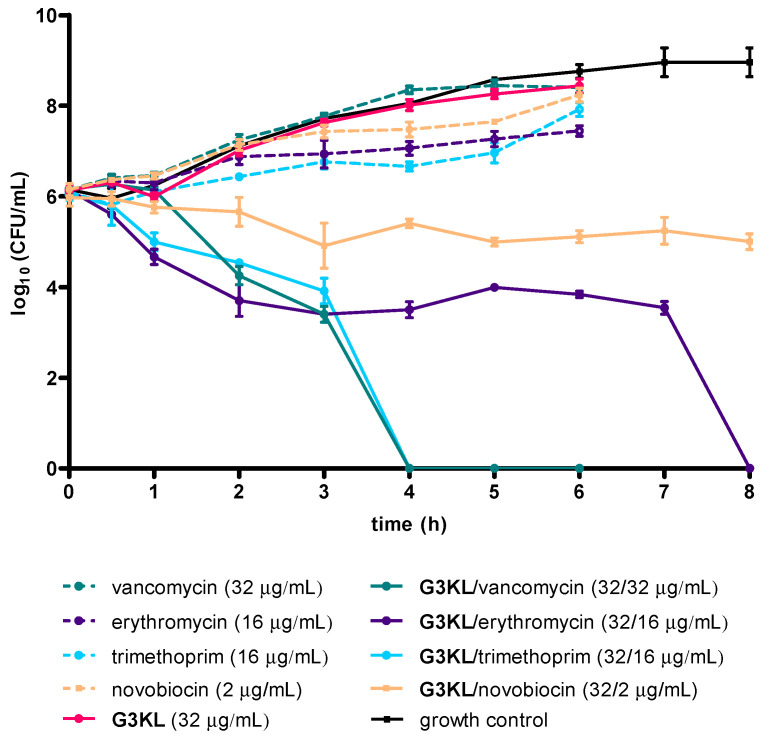
Static time-kill assay with **G3KL**, the antibiotics vancomycin, erythromycin, and novobiocin, and their combination with **G3KL**. The experiment showed a decline in *K. pneumoniae* bacterial burden at 37 °C for the combination of **G3KL**/vancomycin (32 μg/mL/32 μg/mL), **G3KL**/erythromycin (32 μg/mL/16 μg/mL), and **G3KL**/trimethoprim (32 μg/mL/16 μg/mL) below the MIC level. The combination **G3KL**/novobiocin (32 μg/mL/2 μg/mL) showed inhibition in *K. pneumoniae* growth. The assays were performed in triplicate.

**Table 1 molecules-25-05643-t001:** Activity of **G3KL** and **PMB** in combination with **PI**.

	*P. aeruginosa* PAO1	*E. coli* W3110	*A. baumannii* ATCC 19606	*K. pneumoniae* NCTC 418	MRSA COL
**G3KL** ^a^	4	4	8	>64	>64
**PI** ^a^	≥500	62.5–250	>500	≥500	62.5
**PMB** ^a^	0.25	0.25	0.125	0.125	64
**G3KL**_comb_/**PI**_comb_ (FIC_i_) ^b^	2/8 (0.5)	2/31.3 (0.6)	4/15.6 (<0.53)	>64/>500 (-)	>64/62.5 (>2)
**PMB**_comb_/**PI**_comb_ (FIC_i_) ^b^	0.125/62.5 (0.6)	0.125/31.25 (0.6)	0.063/125 (<0.8)	2/16 (>4)	16/31.25 (0.8)

^a^ Minimal inhibitory concentration (MIC) values in μg/mL were determined by serial ½ dilution in MHB (Mueller-Hinton broth). ^b^ MIC in combination in µg/mL was determined by checkerboard method. FIC_i_ was calculated as the sum of FIC of drug A (FIC A) and FIC of drug B (FIC B): FIC_A_(MIC_Acomb_/MIC_A_) + FIC_B_(MIC_Bcomb_/MIC_B_) = ΣFIC = FIC_index_ (FIC_i_). Interpretation of FIC_i_ was as follows: synergistic effect for FIC_i_ < 0.5; partial synergy for 0.5 ≤ FIC_i_ < 1; additive for FIC_i_ = 1; indifferent for 1 < FIC_i_ < 4; antagonism for FIC_i_ ≥ 4 [43]. In the cases where no discrete MIC value was determined in the checkerboard assay, FIC values were calculated by using the highest dilution used in the assay.

**Table 2 molecules-25-05643-t002:** MICs (μg/mL) of **G3KL** and small molecule drugs in combination ^a^.

	*P. aeruginosa* PAO1	*K. pneumoniae* NCTC418	MRSA COL
**G3KL**	4–8	>64	>64
Vancomycin	256	256	0.5
**G3KL**_comb_/Vancomycin_comb_ (FIC_i_) ^b^	1/32 (0.6)	16/16 (<0.3)	>64/0.5 (>2)
Erythromycin	128	64	0.5
**G3KL**_comb_/Erythromycin_comb_ (FIC_i_) ^b^	2/64 (1)	8/8 (<0.3)	32/0.25 (<1)
Ampicillin	>256	>256	128
**G3KL**_comb_/Ampicillin_comb_ (FIC_i_) ^b^	4/>256 (>2)	>64/>256 (-)	32/64 (<1)
Novobiocin	>256	16	0.31
**G3KL**_comb_/Novobiocin_comb_ (FIC_i_) ^b^	2/256 (<1.5)	8/1 (<0.2)	>64/0.31 (>2)
Ciprofloxacin	0.125	0.031	0.25
**G3KL**_comb_/Ciprofloxacin_comb_ (FIC_i_) ^b^	4/0.125 (2)	>32/0.031 (>2)	>64/0.25 (>2)
Chloramphenicol	8	8	8
**G3KL**_comb_/Chloramphenicol_comb_ (FIC_i_) ^b^	2/1 (0.6)	16/2 (<0.5)	>64/8 (>2)
Gentamicin	1	2	0.5
**G3KL**_comb_/Gentamicin_comb_ (FIC_i_) ^b^	4/1 (2)	32/0.25 (<0.6)	64/0.25 (<1.3)
Azithromycin	64	4	4
**G3KL**_comb_/Azithromycin_comb_ (FIC_i_) ^b^	8/0.5 (0.5)	8/1 (<0.4)	16/2 (<0.8)
Sulfamethoxazole	256	>256	>32
**G3KL**_comb_/Sulfamethoxazole_comb_ (FIC_i_) ^b^	1/128 (1)	>64/>256 (-)	>64/>32 (-)
Trimethoprim	128	>256	>32
**G3KL**_comb_/Trimethoprim_comb_ (FIC_i_) ^b^	4/128 (2)	32/8 (<0.5)	64/32 (<1.5)

^a^ The minimal inhibitory concentration in μg/mL was determined by two-fold serial dilutions in MH medium. The experiments were performed in triplicate and the values in µg/mL were calculated based on the peptide mass without trifluoroacetate counterions. ^b^ MIC in combination (**G3KL**/antibiotics) in µg/mL were determined by the checkerboard method. The FIC_i_ in brackets was calculated as the sum of FIC of drug A (FIC A) and FIC of drug B (FIC B): FIC_i_ was calculated as the sum of FIC of drug A (FIC A) and FIC of drug B (FIC B). FIC_A_(MIC_Acomb_/MIC_A_) + FIC_B_(MIC_Bcomb_/MIC_B_) = ΣFIC = FICindex (FIC_i_). Interpretation of FIC_i_ as follows: synergistic effect for FIC_i_ < 0.5; partial synergy for 0.5 ≤ FIC_i_ < 1; additive for FIC_i_ = 1; indifferent for 1 < FIC_i_ < 4; antagonism for FIC_i_ ≥ 4 [43]. In the cases where no discrete MIC value was determined in the checkerboard assay, FIC values were calculated by using the highest dilution used in the assay.

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
