# Peer review of "Synergistic Effect of Propidium Iodide and Small Molecule Antibiotics with the Antimicrobial Peptide Dendrimer G3KL against Gram-Negative Bacteria"

_molecules, 2020, doi:10.3390/molecules25235643_

Round 1

Reviewer 1 Report

The work proposed by Bee-Ha Gan and colleagues, related to G3KL particle and published for the first time in 2014, appears interesting and inserted in a very current research field, namely that of finding valid alternatives to the worrying bacterial resistance to traditional antibiotic drugs. However, in order to give greater prominence to the data presented, some clarifications and, possibly, further details on some important aspects are necessary 

-After defining the FICi indices and ranges, the authors state that for E.coli and A.baumanni there is a "significant synergy" between PI and G3KL, but the values ​​reported in the main text (FICi = 0.6, for E.coli and FICi < 0.53 for A. baumannii) do not fall  within the above definitions, where synergistic effect is defined for FICi < 0.5; and partial synergy for 0.5 ≤ FICi < 1.  Correctly in the legend of the figures S1 and S2, in fact, it is stated that those FICi values ​​indicate a "partial synergy". Authors should  correct the inconsistency in the text

-In relation to the IS, in the legends of all figures ranging from S1 to S5, Cipro is mentioned. However, the antibiotic has not been tested in those experiments, at least as written in the main text. The authors should clarify the possible inconsistency. Furthermore, again in SI, in the experiments shown in Figures S1 to S16, it seems that the MTT dye was used. In that case, no explanation of such a dye was given in any of the legends of the figures.  On the other hand, the meaning and the reason for the use of the MTT dye was never clearly explained in the main text, but only mentioned in the legend of Figure 2 and in the materials and methods.

-The results obtained with the checkerboard technique, which indicate a significant synergy between G3KL  and several antibiotics on K.pneumoniae, are really exciting. However, as the authors themselves know - also on the basis of the references on the synergy between antibiotics cited - the synergies obtained with this technique must be verified and confirmed with more adequate experiments, such as, for example, the time killing curves, widely used for this purpose by the scientific community. Given the relevant results obtained on this important pathogen, I recommend to further verify in this way especially the cases in which the FICi indices indicated a “strong synergy” between G3KL and the selected antibiotic.

Author Response

Reviewer 1 :

The work proposed by Bee-Ha Gan and colleagues, related to G3KL particle and published for the first time in 2014, appears interesting and inserted in a very current research field, namely that of finding valid alternatives to the worrying bacterial resistance to traditional antibiotic drugs. However, in order to give greater prominence to the data presented, some clarifications and, possibly, further details on some important aspects are necessary

-After defining the FICi indices and ranges, the authors state that for E.coli and A.baumanni there is a "significant synergy" between PI and G3KL, but the values ​​reported in the main text (FICi = 0.6, for E.coli and FICi < 0.53 for A. baumannii) do not fall  within the above definitions, where synergistic effect is defined for FICi < 0.5; and partial synergy for 0.5 ≤ FICi < 1.  Correctly in the legend of the figures S1 and S2, in fact, it is stated that those FICi values ​​indicate a "partial synergy". Authors should correct the inconsistency in the text

Our Answer: Thank you very much, we have now corrected to “partial synergy”:

“Partial synergy also occurred with E. coli (FICi = 0.6, Figure S1) and A. baumannii (FICi < 0.53, Figure S2). These data suggest that G3KL permeabilizes the bacterial membranes below its own MIC value to facilitate the entry of PI.”

-In relation to the IS, in the legends of all figures ranging from S1 to S5, Cipro is mentioned. However, the antibiotic has not been tested in those experiments, at least as written in the main text. The authors should clarify the possible inconsistency.

Our answer: For the legend of figures S1 to S5, it was a mistake in the legend, it is now corrected, thank you.

Furthermore, again in SI, in the experiments shown in Figures S1 to S16, it seems that the MTT dye was used. In that case, no explanation of such a dye was given in any of the legends of the figures.  On the other hand, the meaning and the reason for the use of the MTT dye was never clearly explained in the main text, but only mentioned in the legend of Figure 2 and in the materials and methods.

Our answer: We use MTT to reveal the viable bacteria. As the conversion of the dye is spontaneous, we believe the dye does not interfere with the results. This method has been previously reported in our group. It is now incorporated in the main text. Thank you very much.

We have now added in the main text as follow:

To test the feasibility of this approach and possible synergy between antimicrobials and G3KL, we used the classical checkerboard assay and stained live bacteria with 3-(4,5-dimethylthiazol-2-yl)-2,5-Diphenyltetrazolium (MTT) (Figure 2) [31,38,39]

-The results obtained with the checkerboard technique, which indicate a significant synergy between G3KL and several antibiotics on K.pneumoniae, are really exciting. However, as the authors themselves know - also on the basis of the references on the synergy between antibiotics cited - the synergies obtained with this technique must be verified and confirmed with more adequate experiments, such as, for example, the time killing curves, widely used for this purpose by the scientific community. Given the relevant results obtained on this important pathogen, I recommend to further verify in this way especially the cases in which the FICi indices indicated a “strong synergy” between G3KL and the selected antibiotic.

Our answer: Thank you very much for the advises, we have now confirmed the data with time-kill experiments (Figure 3).

Reviewer 2 Report

Gan and coauthors investigated the synergistic effect of the AMP G3KL with the DNA dye propidium iodide and other antibiotics. They found a slight synergistic effect of G3KL with other molecules against P. aeruginosa and a significant synergistic effect against K. pneumoniae. No synergistic effect was observed against Gram positive MRSA.
Besides the fact that the research is very descriptive and does not investigate the specific action mechanisms behind the observed effects, experiments are well conducted and results are sound. I particularly appreciate the decision of showing plate pictures in Supplementary materials.
My only concern is that it is not clear whether experiments were conducted in replicates, especially because it seems that results are different among independent experiments (for example MIC of PI against P. aeruginosa is different in Figure 2 and in Figure S5). Since results were so variable among different cultures/molecule dilutions, performing experiments in triplicates would be needed to confirm the observed results.
Other minor comments:
- I suggest a language revision, as some parts are not very clear;
- Introduction: could authors specify which is the origin of the G3KL peptide?
- Introduction: it is not clear to me what “accumulates up to 10% of the bacterial weight in Gram-negative bacteria” means. Is it the peptide that accumulates weight?
- Figure S1-S5 legends say that the assays show the combination effect of G3KL with PMB and Cipro, but in the text only G3KL and PMB are mentioned. What has been tested?
- In results and discussion, end of Page 4 - beginning of Page 5: a reference is missing about the cited study.

Author Response

Reviewer 2 :

Gan and coauthors investigated the synergistic effect of the AMP G3KL with the DNA dye propidium iodide and other antibiotics. They found a slight synergistic effect of G3KL with other molecules against P. aeruginosa and a significant synergistic effect against K. pneumoniae. No synergistic effect was observed against Gram positive MRSA.

Besides the fact that the research is very descriptive and does not investigate the specific action mechanisms behind the observed effects, experiments are well conducted and results are sound. I particularly appreciate the decision of showing plate pictures in Supplementary materials.

My only concern is that it is not clear whether experiments were conducted in replicates, especially because it seems that results are different among independent experiments (for example MIC of PI against P. aeruginosa is different in Figure 2 and in Figure S5).

Since results were so variable among different cultures/molecule dilutions, performing experiments in triplicates would be needed to confirm the observed results.

Our answer: The number or repetitions has been added in the SI, typically absence of synergy was done once, weak synergies twice, and strong synergies in triplicates and now confirmed with time-kill experiments - with a new Figure 3 and a paragraph describing the time-kill results.

Other minor comments:

- I suggest a language revision, as some parts are not very clear;

- Introduction: could authors specify which is the origin of the G3KL peptide?

Our answer: we have extended the introduction to better explain the origin of G3KL as follows: “G3KL is an antimicrobial peptide dendrimer discovered by optimizing an initial combinatorial library hit[31] by sequence design, and exhibiting remarkable activity...”

- Introduction: it is not clear to me what “accumulates up to 10% of the bacterial weight in Gram-negative bacteria” means. Is it the peptide that accumulates weight?

Our answer: the sentence has been rewritten to be clearer as follows: “and accumulates in Gram-negative bacteria up to an amount of dendrimer corresponding to 10% of the bacterial weight”

- Figure S1-S5 legends say that the assays show the combination effect of G3KL with PMB and Cipro, but in the text only G3KL and PMB are mentioned. What has been tested?

Our answer: Figures S1-S5 show the synergy between PI and either G3KL or PMB, this has been corrected by removing the mention of the experiment involving PI and cipro (which shows no synergy, this data was finally not shown).  

- In results and discussion, end of Page 4 - beginning of Page 5: a reference is missing about the cited study.

Our answer: Thank you for noticing, the missing reference has been added.

Reviewer 3 Report

I found this manuscript to be interesting but somewhat lacking in depth and discussion.  The key peptide dendrimer clearly has some antimicrobial activity(due to membrane disruptive activity), and some additive/synergistic effects are observed with some antibiotics.  However the paper is sorely lacking in any discussion of toxicity associated with either the peptide of the combinations with the various other entities.  Also the discussion of the various results obtained in combination with various other antimicrobials/antibiotics is cursory at best, and there is quite a bit of speculation as to why certain combinations do and don't work but little solid experimental backing for the speculation.  All in all the manuscript is well written and somewhat interesting but I found myself looking or more depth and more analysis of the results. In the end it feels simply like a recitation of the observed results with little specific analysis or explanation of the observations - it lacks depth and simply reads like a recitation of results, more like a laboratory report than a scientific publication, with too much speculation and not enough experimental support or analysis around the conclusions.

Author Response

Reviewer 3:

I found this manuscript to be interesting but somewhat lacking in depth and discussion.  The key peptide dendrimer clearly has some antimicrobial activity(due to membrane disruptive activity), and some additive/synergistic effects are observed with some antibiotics.  However the paper is sorely lacking in any discussion of toxicity associated with either the peptide of the combinations with the various other entities.  Also the discussion of the various results obtained in combination with various other antimicrobials/antibiotics is cursory at best, and there is quite a bit of speculation as to why certain combinations do and don't work but little solid experimental backing for the speculation.  All in all the manuscript is well written and somewhat interesting but I found myself looking or more depth and more analysis of the results. In the end it feels simply like a recitation of the observed results with little specific analysis or explanation of the observations - it lacks depth and simply reads like a recitation of results, more like a laboratory report than a scientific publication, with too much speculation and not enough experimental support or analysis around the conclusions.

Our answer: Here we reported the observation of synergistic effects, which broaden the spectrum of G3KL, with the wellcome addition of Klebsiella pneumoniae against which G3KL is not active. The observation of membrane permeabilization by our peptide dendrimer at sub-inhibitory concentration is not trivial, as evidenced by the facts that 1) not every antibiotic/dendrimer combination leads to synergy, and 2) not every bacterium shows the same sensitivity.  These observations contribute to a better understanding of the mechanism of action of our dendrimer. We believe that this data is significant and of interest for publication.

Round 2

Reviewer 1 Report

Dear Authors,

the new experiments carried out and the supplementary sentences inserted in the text of the article have definitely improved the quality of the manuscript which, in this new revised form, deserves to be published.

with regard

Author Response

This reviewer recommends publication as is. Thank you very much for your support. 

Reviewer 2 Report

Figure legends of Supplementary material still indicate "Cipro" as tested molecule.

Please revise the materials and methods section of the time-kill assay as it presents some inaccuracies (bacterial names not in italics) and grammar errors.

Author Response

 Thank you very much for pointing these typos, the supplementary materials and materials and methods have now been corrected. 

Reviewer 3 Report

The revised version of this manuscript has been improved by the inclusion of kill data for various combinations.  I still don't believe that there is unexpected significance for the combination of an obviously Lys-Arg rich membrane active dendrimer with various antibiotics, but I do think the selectivity for Gm (-) bacteria is interesting and would like to see more experiments with the dendrimer on Gm (-) bacterial membrane models vs other membranes to confirm this .  Also, I think some assessment of the lytic activity/toxicity in mammalian cells is necessary and this must  be added or directly referenced in the manuscript for it to be publishable. I struggle with this manuscript because it is interesting but is not the dramatic breakthrough that is implied.  The authors have added some additional data to the revision and I do think they have made an effort to improve the manuscript , but I struggle whether it is enough. If the lysis/ toxicity data or reference is added, I think the manuscript would then meet the minimum requirements for publication in the journal.

Author Response

Our answers: Thank you very much for your suggestions to better reference the activity of G3KL. The selectivity of G3KL for bacterial vs. mammalian membrane models has been tested previously with vesicle leakage assays as well as its toxicity toward mammalian cells. The text has been modified as follows to better introduce the known properties of G3KL:

G3KL selectively disrupts bacterial versus mammalian membrane models as evidenced by vesicle leakage assays [32], displays pro-angiogenic properties in biological burn-wound bandages [34], anti-biofilm activity [35,36], low toxicity to mammalian cells (IC50 ~1000 μg/mL) [37], and low propensity to resistance development [38].